# Prophylactic Femoral Neck Fixation in an Osteoporosis Femur Model: A Novel Surgical Technique with Biomechanical Study

**DOI:** 10.3390/jcm12010383

**Published:** 2023-01-03

**Authors:** Kyeong-Hyeon Park, Chang-Wug Oh, Joon-Woo Kim, Hee-Jun Kim, Dong-Hyun Kim, Jin-Han Lee, Won-Ki Hong, Jong-Keon Oh

**Affiliations:** 1Department of Orthopedic Surgery, School of Medicine, Kyungpook National University, Kyungpook National University Hospital, Jung-gu, Daegu 41944, Republic of Korea; 2Department of Orthopedic Surgery, School of Medicine, Korea University Guro Hospital, Seoul 10408, Republic of Korea

**Keywords:** femur, nail, osteoporosis, prophylaxis

## Abstract

Intramedullary nailing (IMN) is a popular treatment for elderly patients with femoral shaft fractures. Recently, prophylactic neck fixation has been increasingly used to prevent proximal femoral fractures during IMN. Therefore, this study aimed to investigate the biomechanical strength of prophylactic neck fixation in osteoporotic femoral fractures. An osteoporotic femur model was created to simulate the union of femoral shaft fractures with IMN. Two study groups comprising six specimens each were created for IMN with two standard proximal locking screws (SN group) and IMN with two reconstruction proximal locking screws (RN group). Axial loading was conducted to measure the stiffness, load-to-failure, and failure modes. There were no statistically significant differences in stiffness between the two groups. However, the load-to-failure in the RN group was significantly higher than that in the SN group (*p* < 0.05). Femoral neck fractures occurred in all specimens in the SN group. Five constructs in the RN group showed subtrochanteric fractures without femoral neck fractures. However, one construct was observed in both subtrochanteric and femoral neck fractures. Therefore, prophylactic neck fixation may be considered an alternative biomechanical solution to prevent proximal femoral fractures when performing IMN for osteoporotic femoral fractures.

## 1. Introduction

Intramedullary (IM) nailing (IMN) is the preferred treatment method for femoral diaphyseal fractures in adults. As the elderly population increases, the selection of implants for the fixation of femoral shaft fractures in patients at a high risk of future fractures may be an essential process to ensure satisfactory outcomes. IM nail fixation is the standard treatment for femoral shaft fractures [1,2,3] and is increasing in popularity. However, late femoral neck and proximal peri-implant fractures have been reported after fixation of IMN without femoral neck protection of the femoral shaft fracture in elderly patients [4,5]. Additionally, the severity of osteoporosis is considered a significant risk factor for late hip fractures [4]. Thus, IMN can increase the risk of the femoral neck or proximal peri-implant fractures in elderly individuals with coexisting osteoporosis. This is because it can often be exacerbated by enforced postoperative immobility and a stress riser at the site of the proximal locking screw.

Many surgeons have recently selected a reconstruction nail (RN) with proximally directed interlocking screws to stabilize the femoral shaft and protect the femoral neck when treating femoral shaft fractures in elderly patients with osteoporosis. Several prior studies have investigated the role of prophylactic femoral neck fixation in diaphyseal femoral fractures [4,5,6,7,8] and have suggested that protecting the femoral neck during IM nail fixation of osteoporotic femoral shaft fractures may be effective in reducing late hip fractures [5,6,7]. In addition, a previous biomechanical study demonstrated that IM nail fixation with femoral neck protection can prevent femoral neck fractures [9]. However, it remains unclear whether the same biomechanical difference exists when IMN is performed in osteoporotic femoral models. This study investigated the biomechanical effect of prophylactic neck fixation on the proximal femur during IMN in an osteoporotic femur model.

## 2. Materials and Methods

Twelve synthetic osteoporotic femurs (Model 3503; Pacific Research Laboratories, Vashon, WA, USA) were used in this study. The femurs had a length of 455 mm, a 16 mm hollow canal, and an 18 mm inner cortical diameter. All synthetic femurs were stabilized with antegrade femoral nails (Expert A2FN, DePuy Synthes, Paoli, PA, USA) and divided into two groups according to femoral neck fixation. Six femurs were stabilized using two proximal standard interlocking screws (one oblique and one transverse screw) and two distal interlocking screws (SN group). The other six femurs were stabilized using two proximal reconstruction interlocking screws and two distal interlocking screws (RN group). A synthetic osteoporotic femur was used to simulate a femur with osteoporosis. Osteotomy was not performed after IM nail fixation in the femoral shaft fracture to ensure union.

### 2.1. Specimen Preparation

Each bone model was prepared according to the surgical techniques provided by the implant manufacturer. The nail was inserted at the tip of the greater trochanter (GT) in the anteroposterior (AP) view and parallel with the axial direction of the medullary cavity. The length of the nail was chosen such that its proximal end would meet the GT and its distal end could be placed in the supracondylar area of the distal femur. The length of the nail in the medullary cavity was 380 mm. Because the nail thickness was 10 mm, which was 2 mm less than the pre-measured thickness of the medullary cavity, the nail was inserted after reaming.

In the SN group, a 68 mm-long proximal locking screw was inserted in a 120° antegrade direction until it reached the cortex on the opposite side, for bicortical and firm fixation. Subsequently, a 50 mm-long locking screw was inserted into the static hole of the nail in the transverse direction. Both screws were 5.0 mm thick. In the RN group, two reconstruction screws were inserted and passed through the proximal and distal one-third of the femoral neck in the AP view and through the center of the lateral view. The screw length was chosen to match the distance to the subcortical bone of the femoral head. These hip screws had a thickness of 6.5 mm and lengths of 95 mm and 90 mm for the proximal and distal regions, respectively. After fixation of the proximal interlocking screws, two screws were inserted into the static hole in the distal nail region using a radiation amplifier in both groups. A single orthopedic surgeon performed all the procedures under fluoroscopic guidance to achieve a constant model for the biomechanical study. Proper implantation was confirmed on radiography after instrumentation (Figure 1).

### 2.2. Mechanical Loading

A load was applied to the head of the femur using a custom mold. Each potted femur was placed in a material testing machine (Electroplus E10000, Instron, Norwood, MA, USA) and held with custom fixation devices at 6° valgus to simulate anatomical positioning with weight bearing. The specimens were supported in the testing machine by a ball bearing to avoid uncontrolled torque or bending, as previously described by Cordey et al. [10]. Each distal femur was firmly held in a pre-shaped auto-polymerized acrylic resin (Vertex Dental, Zeist, Netherlands) until the lateral and medial condyles were in contact with the mold as it hardened. The mold aimed to evenly distribute the axial force applied during the testing process (Figure 2).

The experiments were designed to measure the structural stiffness, failure load, and failure mode of each proximal interlocking screw construct by applying axial compression. Before conducting the experiments, an axial preload of 100 N was applied to the servohydraulic testing machine and the femoral constructs to obtain stable results. The test was then performed by loading a 1500 N weight (twice the force applied to the femoral head of a 75 kg adult) at a velocity of 10 N/s in the direction of axial compression, as described by Grisell et al. [11]. This process was repeated five times, and the weight was loaded at 10 N/s on each femur until compressive failure. All constructs were ramped to failure by increasing the force to 10 N/s, and the load, displacement, and time data were collected at a sampling rate of 20 Hz. The axial displacements from the initial position to the preload and from the preload to the maximum load were continuously recorded using the crosshead motion sensor of the servohydraulic testing machine. The degree of displacement corresponding to the increase in axial load was determined for each femur. The stiffness of each femur was calculated as the slope of the elastic portion of the force versus displacement curve, and the mean value was considered. Failure was defined as screw breakage or fracture of a part of the construct, and the force applied to the femoral head at the time of failure was measured. If none of the aforementioned failure criteria were satisfied through direct observation, a sudden reduction in force, as revealed by the force versus displacement graph, was considered a failure [12].

### 2.3. Statistics

Independent sample *t*-tests were used to determine significant differences in stiffness, displacement, and mean failure load between screw constructs. A nonparametric alternative (Mann–Whitney U-test) was used if the hypothesis did not satisfy the parametric method. SPSS Statistics for Windows, Version 18.0 (SPSS Inc., Chicago, IL, USA), was used for statistical analyses, and statistical significance was set at *p* < 0.05.

## 3. Results

### 3.1. Stiffness and Load-to-Failure

There were no gross failures of any construct during or after the repeated cyclic load tests. The mean stiffness in the RN group was 8% higher than that in the SN group. However, there were no statistically significant differences between the two groups. The load-to-failure in the RN group was 25% higher than that in the SN group, and this difference was statistically significant. The descriptive data are presented in Table 1.

### 3.2. Mode of Failures

All constructs in the SN group failed similarly and resulted in basicervical femoral neck fractures from the GT nail entry hole through the proximal oblique interlocking screw (Figure 3A). In the RN group, the five constructs failed without any femoral neck fractures; two constructs sustained subtrochanteric fractures through the reconstruction screw hole without any femoral neck fractures, whereas three constructs sustained subtrochanteric fractures with a non-displaced fracture line that extended toward the GT entry holes. One construct was observed to have subtrochanteric and basicervical femoral neck fractures (Figure 3B).

## 4. Discussion

In this study, we evaluated the mechanical properties of prophylactic femoral neck fixation using reconstruction interlocking screws for IMN in an osteoporotic femur model. Our composite bone model simulated a healed femoral shaft fracture fixed with an IM nail for osteoporosis. Our findings revealed that the reconstruction interlocking constructs showed a higher load-to-failure and prevented delayed femoral neck fractures compared with the standard interlocking construct in an osteoporotic femur model.

Surgeons prefer prophylactic neck fixation for treating femoral fractures. Several authors have advocated prophylactic femoral neck fixation for all femoral shaft fractures because of concerns regarding iatrogenic or missed femoral neck fractures [6,7]. Patton et al. [4] reported late femoral neck fractures after IM nail fixation of femoral shaft fractures in elderly patients. Fourteen patients (2.7%) developed a proximal femoral fracture adjacent to an IM implant in their series. Among them, 11 fractures occurred within months to years. Most patients were > 60 years of age and had osteoporosis and low-energy injuries. Therefore, we suggest that prophylactic neck fixation during the construction of osteoporotic femoral shaft fractures with a reconstruction nail is necessary. Bögl et al. [5] reviewed 897 patients treated for low-energy diaphyseal femoral fractures in Sweden. In their study, 640 patients were treated with IM nails with femoral neck fixation, whereas 257 patients were treated without femoral neck protection. The authors found a five-fold decrease in the risk of reoperation for peri-implant fractures and half the risk of major reoperation when treated with femoral neck fixation. Our study also demonstrated biomechanically that prophylactic neck fixation could effectively prevent femoral neck fractures after IMN for osteoporotic femoral shaft fractures.

Few biomechanical studies have been conducted on prophylactic neck fixation. Previous biomechanical studies using piriformis entry nails have reported that the load-to-failure was similar, regardless of femoral neck protection [13]. They postulated that this may have resulted from a large entry hole through the piriformis fossa, which created a significant bone defect at the base of the femoral neck. Another study using a GT entry nail with neck protection showed a higher load-to-failure than a piriformis entry nail, similar to that of an intact femur [9]. In the present study, higher load-to-failure values were observed in the RN group. This is the result of the difference between the nail entry portal and composite bone used. For osteoporotic femoral shaft treatment, GT entry nail and prophylactic neck fixation may be considered appropriate.

The location of the entry site contributes to the strength of the femoral neck during fixation. In a cadaveric study, Miller et al. [14] showed that the entry site of the piriformis fossa could significantly weaken the neck. All the specimens in their study sustained basicervical fracture patterns under mechanical loading. Strand et al. [15] conducted a cadaveric study comparing entry portals at either the piriformis fossa or the GT of the cadaver. All femurs of the piriformis fossa entry portals sustained basicervical fractures at the entry site. The piriformis fossa group showed a lower load-to-failure than the GT group. As elderly patients with osteoporosis have a lower bone density around the femoral neck, the piriformis entry portal may pose a higher risk of femoral neck fractures in these patients. When performing nailing for femoral fractures, especially in elderly patients, avoiding the piriformis entry portal is recommended. Therefore, our experimental procedure was conducted using the trochanter entry portal.

Synthetic femurs have been widely used and accepted as a substitute for cadaveric specimens in biomechanics. Recently, a new osteoporotic synthetic femur has been introduced by the increasing osteoporosis population. The wall thickness and bone density were reduced to simulate osteoporotic bone. In biomechanical studies, the osteoporotic synthetic bone shows similar axial loading results compared with the osteoporotic bone [16].

Our study had several limitations. First, this was a biomechanical study using synthetic femur models and, thus, may not accurately represent human bone mechanics, especially in elderly patients. However, synthetic bone provides several advantages over cadaveric bone; thus, it is preferred in biomechanical studies [12,17]. Furthermore, synthetic bones can provide standard sizes and properties between specimens and guarantee implantation techniques’ reproducibility. In addition, cadaveric specimens of varying dimensions, ages, and bone densities were excluded. Moreover, our approach is meaningful because we used a newly developed synthetic osteoporotic bone biomechanically similar to an osteoporotic cadaveric femur [16].

Second, the axial load provided by a mechanical testing machine may not accurately mimic the physiological load or vector experienced during a standard ground-level fall. We assumed that femur neck fracture is a result of axial load through the mechanical axis of the femur. We decided to use this assumption to enhance reproducibility, with the thought that weight bearing goes through the mechanical axis from a functional perspective. Other forces we did not simulate can cause femur neck fractures. However, the testing model used in this study has been validated and previously utilized in multiple biomechanical studies. Consequently, this is an appropriate and applicable method for isolating the strength of the proximal femur.

Third, we only tested load-to-failure rather than cyclic loading. Failure loading simulates a fall or other trauma. Moreover, some clinical reports had either no or minimal trauma due to the mechanism of the fracture, implying that femur neck fracture may be due to repetitive loading, leading to a stress fracture. However, our goal was to investigate the protection the fixation construct provides against a catastrophic event such as a fall.

Finally, this study was conducted only with an osteoporotic bone model, so the effect of prophylactic neck fixation is specific to the osteoporotic femur. Additional samples and comparison studies with non-osteoporotic bone may have provided a more accurate representation of the load-to-failure of the femur models.

## 5. Conclusions

In summary, the results of this biomechanical study showed that femoral nailing with two reconstruction screws resulted in a higher load-to-failure than femoral nailing with standard screws in an osteoporotic femoral model. Clinically, it may be assumed that IM nail fixation using reconstructive screws could prevent femoral neck fractures in patients with osteoporosis.

## Figures and Tables

**Figure 1 jcm-12-00383-f001:**
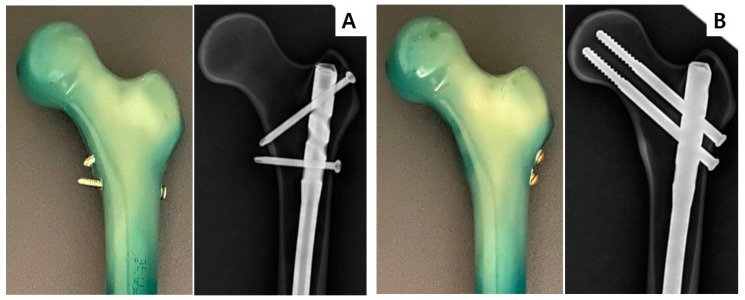
Two different constructs of the osteoporotic femur model with IMN and radiographs: (**A**) IMN with two standard proximal locking screws (SN group) and (**B**) IMN with two reconstruction proximal locking screws (RN group). IMN, intramedullary nailing.

**Figure 2 jcm-12-00383-f002:**
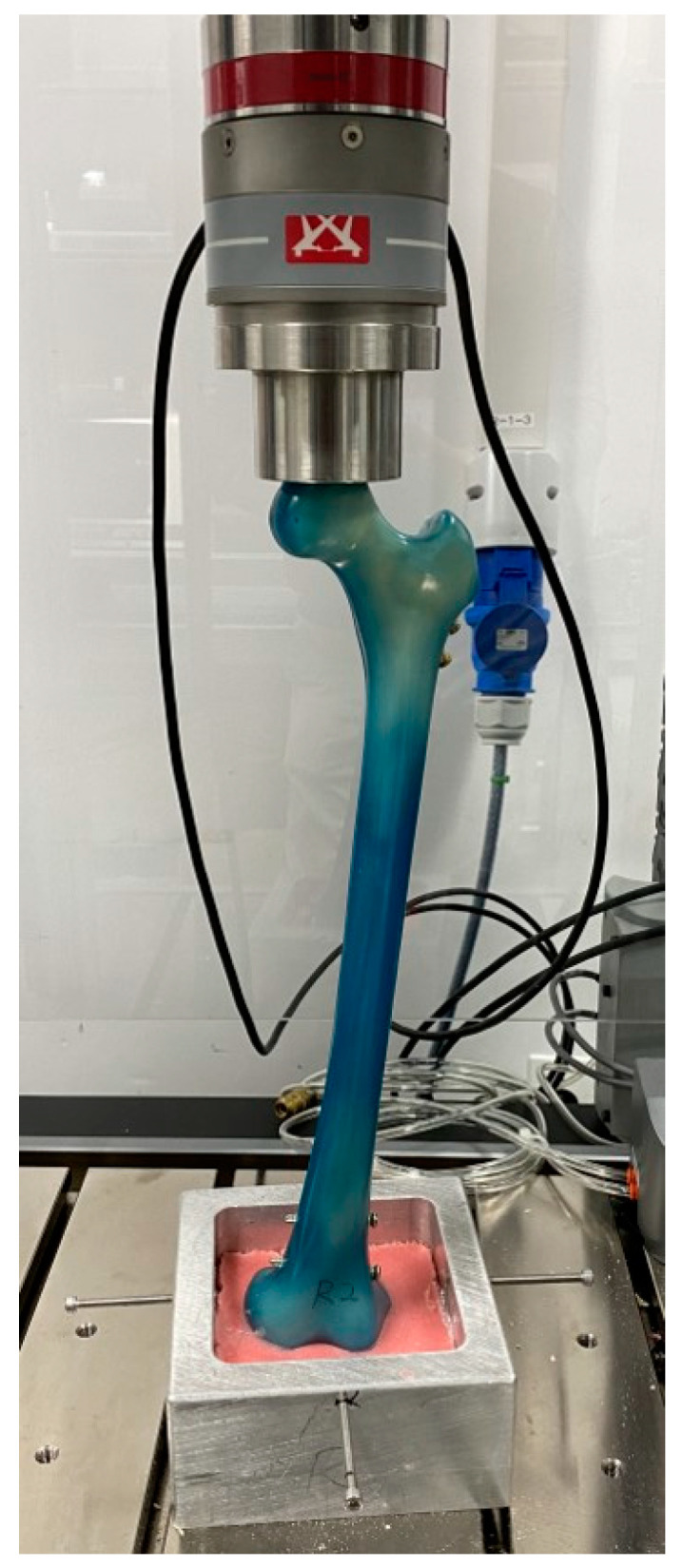
Experimental setup for measuring the stiffness and failure load after creating the osteoporotic femur models with different proximal interlocking screw constructs.

**Figure 3 jcm-12-00383-f003:**
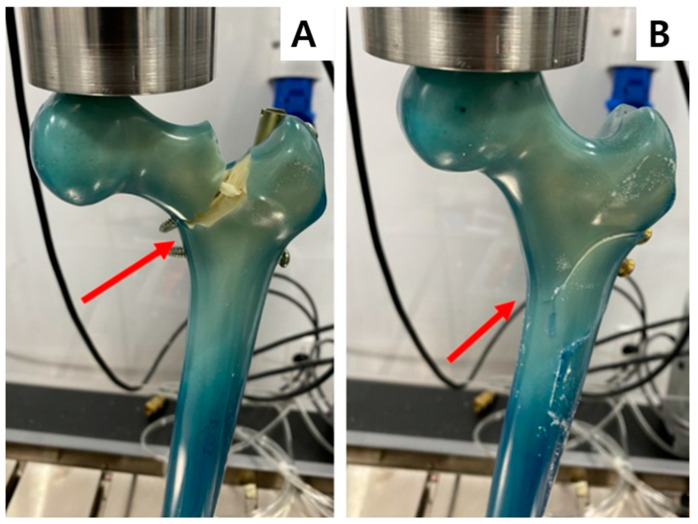
Failure modes after axial compressive loads. All constructs of group SN showed a basicervical femoral neck fracture from the GT nail entry hole through the proximal oblique interlocking screw (**A**). In the RN group reconstruction nail, five constructs exhibited a subtrochanteric fracture through the reconstruction screw hole, but without a femoral neck fracture (**B**). GT, greater trochanter. Red arrows point to the fracture lines.

**Table 1 jcm-12-00383-t001:** Measurement results of construct stiffness under axial compressive load.

Construct	Axial Compressive Load	
	Stiffness (N/mm)	Load-to-Failure (N)
IM nail with standard interlocking screws (SN Group)	506 ± 37	2426 ± 163
IM nail with reconstruction interlocking screws (RN Group)	545 ± 57	3020 ± 103
*p*-value	0.31	<0.05

## Data Availability

Source data may be shared upon reasonable request to the corresponding author.

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
