# Peer review of "Prophylactic Femoral Neck Fixation in an Osteoporosis Femur Model: A Novel Surgical Technique with Biomechanical Study"

_jcm, 2023, doi:10.3390/jcm12010383_

Round 1

Reviewer 1 Report

Dear Authors,

Hope you are well. I thank you teh opportunity to read your work “Prophylactic femoral neck fixation in an osteoporosis femur 2 model: A novel surgical technique with biomechanical study”, submitted for publication at JCM. The article raises an important issue related to prophylactic neck fixation as an alternative biomechanical solution to prevent proximal femoral fractures when performing IMN for osteoporotic femoral fractures. The article is a very interesting manuscript, well designed, written, and easy to be followed by the reader.

Best regards

Author Response

Response to Reviewer 1 Comments

Dear Authors,

Hope you are well. I thank you the opportunity to read your work “Prophylactic femoral neck fixation in an osteoporosis femur 2 model: A novel surgical technique with biomechanical study”, submitted for publication at JCM. The article raises an important issue related to prophylactic neck fixation as an alternative biomechanical solution to prevent proximal femoral fractures when performing IMN for osteoporotic femoral fractures.

Point 1: The article is a very interesting manuscript, well designed, written, and easy to be followed by the reader.

Best regards

Response 1: Thank you. The authors are very grateful for your comment.

Reviewer 2 Report

This paper compares the biomechanical characteristics of the two different techniques using the osteoporosis bone model. Few experimental data are available to conclude the usefulness of prophylactic neck fixation, and additional experiments should be conducted. 

There are few tables and figures showing experimental results, which are difficult to understand.

It is not clear why the authors can conclude that prophylactic neck fixation is effective with the methodology of this study.

Is this enough of a variety of biomechanical tests?

What about the experimental results using the normal bone model instead of the osteoporosis bone model?

Author Response

Response to Reviewer 2 Comments

Point 1: This paper compares the biomechanical characteristics of the two different techniques using the osteoporosis bone model. Few experimental data are available to conclude the usefulness of prophylactic neck fixation, and additional experiments should be conducted.

Response 1: We agree with your opinion. Clinically, it has been reported that femur neck fracture frequently occurs after IM nailing of the femur in elderly patients with osteoporosis. However, to the authors' knowledge, the significance of this study is that it is the first biomechanical study on prophylactic neck fixation in an "osteoporosis femur model." Although this study alone can not conclude the usefulness of prophylactic neck fixation, it can provide theoretical evidence for future studies. We added a comment to the limitation.

“Finally, this study was conducted only with an osteoporotic bone model, so the effect of prophylactic neck fixation is specific to the osteoporotic femur. Additional samples and comparison studies with non-osteoporotic bone may have provided a more accurate representation of the load-to-failure of the femur models.”

Point 2: There are few tables and figures showing experimental results, which are difficult to understand. It is not clear why the authors can conclude that prophylactic neck fixation is effective with the methodology of this study.

Is this enough of a variety of biomechanical tests?

Response 2: There are various methods of biomechanical testing, such as axial loading, axial rotation, lateral bending, and flexion-extension, which are selected according to the purpose of the study. We assumed that femur neck fracture is a result of axial load through the mechanical axis of the femur in the event of trauma such as a fall. So, we decided to use this assumption to enhance reproducibility, with the thought that weight-bearing goes through the mechanical axis from a functional perspective. The testing model of the proximal femur used in this study has been validated and previously utilized in multiple biomechanical studies (1-3) using an axial loading test. Consequently, this is an appropriate and applicable method for isolating the strength of the proximal femur. However, axial loading could not simulate all fracture situations, which was explained as a limitation of this study.

<Reference>

  1. Shieh AK, Bravin DA, Shelton TJ, Garcia-Nolen TC, Lee MA, Eastman JG. A Biomechanical Comparison of Trochanteric Versus Piriformis Reconstruction Nails for Femoral Neck Fracture Prophylaxis. J Orthop Trauma 2021;35:e293–7.
  2. Shieh AK, Refaat M, Heyrani N, Garcia-Nolen TC, Lee MA, Eastman JG. Are piriformis reconstruction implants ideal for prophylactic femoral neck fixation? Injury 2019;50:703–7.
  3. Crump EK, Quacinella M, Deafenbaugh BK. Does Screw Location Affect the Risk of Subtrochanteric Femur Fracture After Femoral Neck Fixation? A Biomechanical Study. Clin Orthop Relat Res. 2020 Apr;478(4):770-776.

Line: 216-224, “Second, the axial load provided by a mechanical testing machine may not accurately mimic the physiological load or vector experienced during a standard ground-level fall. We assumed that femur neck fracture is a result of axial load through the mechanical axis of the femur. We decided to use this assumption to enhance reproducibility, with the thought that weight-bearing goes through the mechanical axis from a functional perspec-tive. Other forces we have not simulated can cause femur neck fractures. However, the testing model used in this study has been validated and previously utilized in multiple biomechanical studies. Consequently, this is an appropriate and applicable method for isolating the strength of the proximal femur.”

Point 3: What about the experimental results using the normal bone model instead of the osteoporosis bone model?

Response 3: A previous biomechanical study with a normal femur model demonstrated that IM nail fixation with femoral neck protection could prevent femoral neck fractures. The femur neck fractures are increasing in patients with osteoporosis after IM nailing. Prophylactic neck fixation was recently emphasized during surgery for femoral shaft fractures with osteoporosis. However, it remains unclear whether the same biomechanical difference exists when IM nailing is performed in osteoporotic femoral models. Although the experiment was not performed on a normal femur model, this study has the strength of reporting for the first time that prophylactic neck fixation in synthetic osteoporotic bone shows better biomechanical stability. We also agree with your comment, and comparing normal bones together later may be necessary. We described the comment as the limitations.

Line: 230-233, “Finally, this study was conducted only with an osteoporotic bone model, so the effect of prophylactic neck fixation is specific to the osteoporotic femur. Additional samples and comparison studies with non-osteoporotic bone may have provided a more accurate representation of the load-to-failure of the femur models.”

Reviewer 3 Report

The paper presented for the review is aimed to investigating the biomechanical effect of prophylactic neck fixation on the proximal femur during IMN in an osteoporotic femur model. The authors conducted the elegant experimental study with the mechanical models of the osteoporotic femur with two types of implants and several technical approaches for the intramedullary nail implantation. They came to the interesting findings which have possible practical importance for that surgical approach to the provisional fixation of the femoral neck in cases of the femoral shaft fracture. The major recommendation is that femoral nailing with two reconstruction screws resulted in a higher load-to-failure than femoral nailing with standard screws in an osteoporotic femoral model.  Some minor modifications could make the paper more clear from the reviewer’s point of view. The object of the study includes synthetic models. Despite the references in the text it seems more clear if brief description of the essential information regarding the specific features of the model as the “osteoporotic” would presented. It seems also important to notice among the limitations that the authors didn’t provide the comparison with the non-osteoporotic models and it is not actually proved that the findings are specific for the “osteoporotic” bones and not for the femur generally.

Author Response

Response to Reviewer 3 Comments

The paper presented for the review is aimed to investigating the biomechanical effect of prophylactic neck fixation on the proximal femur during IMN in an osteoporotic femur model. The authors conducted the elegant experimental study with the mechanical models of the osteoporotic femur with two types of implants and several technical approaches for the intramedullary nail implantation. They came to the interesting findings which have possible practical importance for that surgical approach to the provisional fixation of the femoral neck in cases of the femoral shaft fracture. The major recommendation is that femoral nailing with two reconstruction screws resulted in a higher load-to-failure than femoral nailing with standard screws in an osteoporotic femoral model.  Some minor modifications could make the paper more clear from the reviewer’s point of view.

Point 1: The object of the study includes synthetic models. Despite the references in the text it seems more clear if brief description of the essential information regarding the specific features of the model as the “osteoporotic” would presented.

Response 1: Thanks for the comment. We have added relevant information to the discussion regarding your comments.

Line 202-207: “Synthetic femurs have been widely used and accepted as a substitute for cadaveric specimens in biomechanics. Recently, a new osteoporotic synthetic femur has been introduced by the increasing osteoporosis population. The wall thickness and bone density were reduced to simulate osteoporotic bone. In biomechanical studies, the osteoporotic synthetic bone shows similar axial loading results compared with the osteoporotic bone [16].”

Point 2: It seems also important to notice among the limitations that the authors didn’t provide the comparison with the non-osteoporotic models and it is not actually proved that the findings are specific for the “osteoporotic” bones and not for the femur generally.

Response 2: Thank you for your comment. A previous biomechanical study with a normal femur model demonstrated that IM nail fixation with femoral neck protection could prevent femoral neck fractures. It has been reported that femur neck fractures are increasing in patients with osteoporosis. However, it remains unclear whether the same biomechanical difference exists when IMN is performed in osteoporotic femoral models. Prophylactic neck fixation has recently been emphasized during surgery for femoral shaft fractures. Although the experiment was not performed on a normal femur model, this study has the strength of reporting for the first time that prophylactic neck fixation in synthetic osteoporotic bone shows excellent biomechanical stability. We also agree with your comment, and comparing normal bones together later is necessary. We described the comment as the limitations.

Line: 230-233, “Finally, this study was conducted only with an osteoporotic bone model, so the effect of prophylactic neck fixation is specific to the osteoporotic femur. Additional samples and comparison studies with non-osteoporotic bone may have provided a more accurate rep-resentation of the load-to-failure of the femur models.”

Round 2

Reviewer 2 Report

This revised manuscript answers the reviewers’ questions and addresses their concerns. It deserves attention and is worthy of publication in this journal.